

# Accelerometry-assessed daily physical activity and compliance with recommendations in Spanish children: importance of physical education classes and vigorous intensity

Juan Carlos Benavente-Marín[1], Francisco Javier Barón-López[1], Begoña Gil Barcenilla[2], Guadalupe Longo Abril[2], José M. Rumbao Aguirre[2], Napoleón Pérez-Farinós[1,3] and Julia Wärnberg[1,3]

[1] EpiPHAAN Research Group, Universidad de Málaga—Instituto de Investigación Biomédica de Málaga (IBIMA), Málaga, Spain
[2] Plan Integral de Obesidad Infantil de Andalucía (PIOBIN), Consejería de Salud y Consumo. Junta de Andalucía, Sevilla, Spain
[3] Centro de Investigación Biomédica en Red Fisiopatología de la Obesidad y la Nutrición (CIBEROBN), Institute of Health Carlos III, Madrid, Spain

Corresponding authors
Napoleón Pérez-Farinós,
napoleon.perez@uma.es
Julia Wärnberg, jwarnberg@uma.es

## ABSTRACT

**Background:** Physical activity (PA) is associated with numerous health benefits. Vigorous PA (VPA) may have a greater impact on public health than lower-intensity PA. The incorporation of a specific recommendation on VPA could complement and improve existing recommendations for average daily moderate-vigorous PA (MVPA). Physical education classes could have a positive impact on children's adherence to average daily physical activity recommendations. The aim was to investigate the association between MVPA and VPA in children, as well as adherence to recommendations, and obesity and the presence of physical education classes.

**Methods:** A cross-sectional study of physical activity was conducted in a sample of 8 and 9-year-old children in Andalusia (Spain). GENEActiv accelerometers were used, placed on the non-dominant wrist for at least eight consecutive days (24-h protocol). School days with and without physical education class, and weekend days were defined. ROC curves were used to calculate the threshold associated with obesity for average daily MVPA and VPA for recommendations.

**Results:** A total of 360 schoolchildren were included in the analyses (184 girls). An average of 7.7 (SD 1.4) valid days per participant were evaluated, with 19.9 (SD 10.5) and 11.4 (SD 5.1) minutes of VPA performed by boys and girls respectively. 25.8% of the participants were classified with central obesity. The optimal threshold determined with ROC analysis was 12.5 and 9.5 minutes of average daily VPA for boys and girls, respectively (RecVPA), and 75 minutes of average daily MVPA for both sexes (RecMVPA). The RecVPA showed stronger association with obesity. On school days with physical education class, compared to days without this class, children showed increased VPA and MVPA engagement and better compliance with recommendations, with smaller differences in adherence according to sex or obesity.

**Conclusions:** On days with physical education class, more physical activity was accumulated at all intensities and greater adherence to the recommendations than on

days without this class. VPA had a stronger correlation with the absence of obesity than lower-intensity activity. It was also observed that boys were physically more active and had higher adherence to the recommendations than girls.

## INTRODUCTION

Physical activity (PA) in children and adolescents is associated with numerous health benefits (*Poitras et al., 2016*), independent of sedentary behavior (*Ekelund et al., 2012*). The World Health Organization's (WHO) 2020 Guidelines on Physical Activity and Sedentary Behavior, emphasizes various benefits of physical activity for children and adolescents, including improved physical fitness, cardiometabolic health, bone health, cognitive outcomes, mental health, and reduced adiposity (*Chaput et al., 2020*). The WHO recommends that children and adolescents should do an average of at least 60 min per day of moderate to vigorous, mostly aerobic, physical activity (MVPA) across the week and should incorporate vigorous-intensity aerobic activities, as well as those that strengthen muscle and bone, on at least 3 days a week (*World Health Organization (WHO), 2020*). However, the recommended time for vigorous activity is not specified, which complicates the objective evaluation of this recommendation.

Consequently, the majority of studies have concentrated on assessing adherence to the MVPA recommendation, and the results have been heterogeneous (*Van Hecke et al., 2016*). Some researchers suggest that current guidelines may underestimate the physical activity necessary to reduce cardiovascular risk (*Füssenich et al., 2016*) and propose that it is necessary to delve into the measurement of vigorous physical activity (VPA), which may have a greater impact on public health than lower-intensity physical activity (*Füssenich et al., 2016*; *Poitras et al., 2016*; *Aadland et al., 2018*), although no specific recommendation exists. Some studies have suggested that accumulating between 10–20 minutes (averaging 15 minutes) of daily VPA may be related to improvements in cardiometabolic markers, adiposity, cardiorespiratory fitness, bone mineral density, and cardiovascular risk in children and adolescents (*Martinez-Gomez et al., 2010*; *Gralla et al., 2016*; *Füssenich et al., 2016*; *Schwarzfischer et al., 2017*; *Larsen et al., 2018*; *García-Hermoso et al., 2021*; *Gammon et al., 2022*). Therefore, incorporating a specific recommendation for VPA could complement and enhance the existing recommendations for MVPA, making them more useful for public health intervention and the prevention of childhood obesity.

Likewise, there is a significant disparity between the sexes in the performance of MVPA, particularly VPA, where boys are consistently observed to be more physically active than girls (*Laguna et al., 2013b*; *Katzmarzyk et al., 2015*; *Telford et al., 2016*; *Gralla et al., 2016*; *Füssenich et al., 2016*; *Guthold et al., 2020*; *Steene-Johannessen et al., 2020*; *Gammon et al., 2022*). Differences in physical activity-related attributes between the sexes have been identified, such as cardiorespiratory fitness, hand-eye coordination, body fat percentage,

and perceived competence in physical education (*Telford et al., 2016*). Girls also encounter specific barriers to physical activity engagement, such as aversion to physical activity and lack of time (*Delfa-De-La-Morena et al., 2022*). It may not be appropriate to seek physical activity recommendations that are equally effective for both sexes. In fact, some authors have proposed different recommendations for boys and girls that optimize their ability to identify those with unfavorable health markers (*Laguna et al., 2013b*; *Katzmarzyk et al., 2015*; *Gralla et al., 2016*; *Schwarzfischer et al., 2017*).

On the other hand, there is a high prevalence of overweight and obesity in children and adolescents (*NCD Risk Factor Collaboration, 2017*; *ALADINO, 2020*; *Bibiloni et al., 2022*; *Serra Majem et al., 2003*), which increases their risk of developing diseases in adulthood (*Umer et al., 2017*; *Migueles et al., 2023*). Therefore, having physical activity recommendations that are as tailored as possible to the biological circumstances of children and adolescents, as well as promoting programs that aid in meeting these recommendations, can play an essential role in preventing excess body weight in this population (*Gralla et al., 2016*; *Poitras et al., 2016*; *Owens, Galloway & Gutin, 2016*; *World Health Organization (WHO), 2020*; *Chaput et al., 2020*; *García-Hermoso et al., 2021*; *Migueles et al., 2023*).

Finally, an aspect that warrants further exploration is the variation in physical activity on different types of day, such as school days *vs* non-school days, and days with physical education classes *vs* those without. In Spain, children and adolescents attend school from September to June, 5 days a week, with at least two of those days including physical education classes. Prior findings suggest that MVPA levels are typically higher on school days, particularly on days with physical education classes (*Brooke et al., 2016*; *Mayorga-Vega, Martínez-Baena & Viciana, 2018*). It is crucial to examine how these different types of day might influence physical activity patterns, particularly VPA, as well as adherence to the recommendations. Furthermore, it is pertinent to investigate potential disparities based on sex and obesity status. Gaining this knowledge will aid in identifying opportunities to enhance compliance with recommendations and in devising specific strategies for each group.

The daily duration and intensity of physical activity could potentially be associated with obesity in children. Furthermore, the presence of physical education classes might influence the likelihood of these children meeting the daily MVPA and VPA recommendations. This association could vary according to sex and obesity status.

The overarching aim of this study is to investigate the association between MVPA and VPA in 8- and 9-year-old children, and obesity and the presence of physical education classes. To achieve this general aim, the study proposes the following specific objectives: (1) To objectively measure the daily duration of MVPA and VPA in 8- and 9-year-old children, calculate thresholds at which daily MVPA and VPA are associated with obesity, and utilize these values in recommendations for obesity prevention. (2) To assess the relationship between the daily duration of MVPA and VPA and days with and without physical education classes, according to sex and obesity. (3) To evaluate the association between adherence to daily MVPA and VPA recommendations and days with and without physical education classes, based on sex and obesity. (4) To estimate the probability of

meeting daily MVPA and VPA recommendations based on the presence or absence of physical education classes, according to sex and obesity.

## MATERIALS AND METHODS

### Study design and sample

We conducted a cross-sectional study to examine physical activity using accelerometry in a representative sample of 8- and 9-year-old children from the ALADINO 2019 study (ALimentación, Actividad física, Desarrollo INfantil y Obesidad in Spain in 2019) in Andalusia, Spain. Andalusia, located in the southern Iberian Peninsula, is one of the 17 Autonomous Communities into which Spain is divided. The ALADINO 2019 study was conducted in Andalusia by the Spanish Agency for Food Safety and Nutrition (AESAN) in collaboration with the Andalusian regional childhood obesity plan PIOBIN (Plan Integral de Obesidad Infantil de Andalucía). In the ALADINO 2019 study, 40 primary education schools in Andalusia participated. Both the design and methodology of the ALADINO 2019 study were developed in accordance with the protocols and recommendations of the WHO European Childhood Obesity Surveillance Initiative (COSI Euro WHO) (*ALADINO, 2020*).

A predetermined minimum sample size was calculated for each of the estimations to be carried out in the study. The estimation that required the largest sample size was the determination of thresholds for VPA that could differentiate between children with and without central obesity. For this calculation, we used an area under the curve (AUC) of 0.64 (*Schwarzfischer et al., 2017*). Additionally, it was estimated that approximately 25% of the study population would have central obesity. Using these data as a reference, with a 95% confidence level ($\alpha = 0.05$) and accepting a power ($1-\beta$) of 0.8, the minimum required sample size was 163 children. It was estimated that about half of the participating children would be boys, and the other half girls, so a minimum sample of 326 children was considered necessary to obtain representative results for both sexes, in addition to the total participants.

An average classroom size of 20 students in 3rd grade of primary education was assumed, with a 20% refusal rate to participate. Therefore, to reach the estimated sample size, we needed to evaluate one 3rd grade classroom in 20 primary education schools. For this current study, all 40 primary education schools that participated in the ALADINO 2019 study in Andalusia were invited to participate, thereby reaching and surpassing the calculated minimum sample size.

Inclusion criteria for this study were: 1) enrollment in the 3rd grade of primary education during the 2019/2020 academic year in a primary education school participating in the ALADINO 2019 study in Andalusia; and 2) having a signed informed consent from legal guardians authorizing participation in the specific accelerometry study. Participants with limitations for physical activity during the evaluation and those over 9.99 years old were excluded from the analysis.

We reported this study as per the Strengthening the Reporting of Observational Studies in Epidemiology (STROBE) guideline (Supplemental STROBE Statement) (*Cuschieri, 2019*). This study is a supplementary study to the ALADINO 2019 study in Andalusia,

incorporating newly collected accelerometer data. The legal guardians of the participants were offered the opportunity to obtain a report of their children's individual results. Similarly, participating schools were offered the opportunity to obtain a report of the average results of the participating student group.

The study was conducted in accordance with the Declaration of Helsinki (*World Medical Association, 2013*) and approved by the research ethics committee CEI-Costa del Sol and the Portal de Ética de la Investigación Biomédica de Andalucía-PEIBA, the 26[th] of September 2019, with the reference number 0114-2019. All data was carried out respecting the European legislation 2016/679 of data protection, and the Spanish 'Organic Law 3/ 2018 of December 2005'. The clinical data was kept segregated and encrypted. The signed informed consent was obtained from all legal guardians.

## Data collection procedure

In the schools that agreed to participate, all 3[rd] grade primary school children from the classroom selected for the ALADINO 2019 study (*ALADINO, 2020*) in Andalusia were invited. The legal guardians of the children received the invitation in written form with all the information about the study.

Data collection was distributed evenly during the 2019–20 school year according to the size of the populations where the participating schools are located. Data collection was planned to be carried out between October 2019 and June 2020, but it ended in March 2020 due to the pandemic caused by the SARS-CoV-2 virus. In Spain, schools were closed from March 15, 2020, until the end of the 2019–20 academic year and home confinement of the population was decreed. For this reason, the results of this study are prior to the COVID-19 pandemic.

Two visits were made to each school. In the first visit, accelerometers were individually placed on the participants. Teachers and participants were asked to maintain their daily activities during the accelerometry evaluation. In the second visit, accelerometers were removed, and each participant was asked if they had removed the accelerometer or if they had missed class during the evaluation. The schedule of the school time and physical education classes on the evaluated days was recorded. All participating schools began at 9:00 AM and ended at 2:00 PM. All had one daily 30-minutes recess around the middle of the school time, starting between 11:00 AM and 12:00 PM.

## Criteria for accelerometry data collection

For the objective evaluation of physical activity, GENEActiv accelerometers (Activinsights Ltd., Kimbolton, UK) were used. These are triaxial accelerometers with a dynamic range of ±8 gravitational units (g), where 1 g equals Earth's gravitational pull. The accelerometers were configured with a sampling frequency of 40 Hz using GENEActiv PC Software (version 3.2).

The accelerometers were worn on the non-dominant wrist, and participants were asked to wear them continuously for at least eight consecutive days to ensure a complete assessment of five school days and a weekend. Participants and their families were instructed not to remove the device at any time during the assessment (24 hours protocol).

It was emphasized that the device was waterproof, and participants were required to wear it while sleeping.

## Processing of accelerometry data

No noise filter was applied prior to processing. Raw accelerometer data files were processed using R (*R Core Team, 2021*) with the R package accelerator (version 0.4.0) (*Barón-Suárez et al., 2023*). The processing included the processing functions of the R GGIR package (version 2.9.2) (*Migueles et al., 2019*). In summary, GGIR performed the following tasks: (1) Auto-calibration (*van Hees et al., 2014*); (2) detection of abnormally high sustained values; (3) non-wear time detection; (4) calculation of the Euclidean norm minus one with negative values set to zero (ENMONZ or ENMO) (*van Hees et al., 2013*). The raw data were simplified by calculating ENMONZ values (measured in milligravitational units, mg) in 5-second epochs (*Baquet et al., 2007*; *Aadland et al., 2018*).

The GGIR algorithm was found to be inadequate in detecting relatively short non-wear periods, so the GGIR non-wear time definition was supplemented with strict periods of sustained inactivity. These periods needed to last at least 30 minutes, with angle changes in the Z-axis below two degrees, calculated between 8:00 AM and 10:00 PM.

To classify physical activity by intensity, the cut-off points published by *Hildebrand et al. (2014*, *2017*) for GENEActiv accelerometers, placed on the non-dominant wrist, in children aged 7 to 11 years, and expressed in ENMONZ (mg) were used (*Hildebrand et al., 2014*, *2017*). The specific cut-off points used were as follows: light physical activity (LPA, from 56.3 to 191.6 mg), moderate physical activity (MPA, from 191.6 to 695.8 mg), VPA (over 695.8 mg), physical activity at any intensity (LMVPA, over 56.3 mg), and MVPA (over 191.6 mg).

When participants reported removing the accelerometer for a known sporting activity, it was checked if this coincided with non-wear time. If confirmed, non-wear time was replaced with mean values for a similar sporting activity, which had been observed and studied in other participants from the same sample.

Four types of day were defined for the analysis: weekly days, school days with physical education class, school days without physical education class, and weekend days. An evaluated day was considered valid when the accelerometer was active and recording for a minimum of 20 hours (from 00:00 to 00:00 h) with no more than 2 hours of non-wear time accumulated between 8:00 AM and 10:00 PM. School time was valid if the accelerometer recorded at least 4 hours with no more than 1 hour of non-wear time during school hours. A physical education class was considered valid if it accumulated less than 1 minute of non-wear time, included at least 3 minutes of MVPA, or did not exceed 30 minutes of sedentary behavior (*i.e.*, epochs with less than 56.3 mg). Physical education classes for non-participating students were excluded.

Consequently, a valid school day implied a valid school time. If a valid school day also included a valid physical education class, it was considered a school day with physical education class. If the valid day was a Saturday or Sunday, it was considered a weekend day. To calculate the weekly day, daily average results were weighted with 5/7 for the average of school days and 2/7 for the average of weekend days. If an assessment had two

identical days of the week (*e.g.*, two Mondays), these were averaged, and this average was used as the mean value for that type of day. An assessment was considered valid when it had at least four valid weekly days (*Antczak et al., 2021*), of which at least two school days and at least 1 weekend day. To maintain precision, holidays and school absence days were excluded from the analysis.

If there was non-wear time in the resulting valid days, it was imputed by the average value of the different intensities of physical activity calculated for the same type of day in the time interval occupied by the non-wear time. If the average value for that same type of day was not available, it was imputed with the average weekly daily value for that time interval.

## Other study variables

Information regarding sex and date of birth was collected in the informed consents. Age was calculated as the difference between the start date of the accelerometry evaluation and the participant's date of birth.

Body weight, height, and waist circumference were measured between October and December 2019. The TANITA model UM-076 scale was used, capable of recording weights from 0 to 150 kg with a precision of 100 g. Heights were measured using the portable SECA model 206 stadiometer, which measures between 0 and 220 cm with a precision of 1 mm. Waist circumferences were measured using the SECA model 201 anthropometric measuring tape, with a measuring range of 0 to 205 cm and a precision of 1 mm. Body mass index (BMI) was calculated as weight divided by height squared (kg/m$^2$). Weight status was classified into three categories (normal weight, overweight, and obesity) using the WHO growth standards (*de Onis et al., 2007*) and the cutoff points of the International Obesity Task Force (IOTF) (*Cole & Lobstein, 2012*). Waist-to-Height Ratio (WHtR) was calculated as waist circumference (cm) divided by height (cm). Children with central obesity were classified as those with a WHtR greater than or equal to 0.5 (*Eslami et al., 2022*).

Information on the highest level of education of the parents (university or non-university studies) and the type of school (public or private) was also collected.

## Statistical analysis

The mean, standard deviation (SD), minimum, maximum, and total valid days were calculated for the four types of day studied. The average time for LPA, MPA, and VPA were calculated for the participants in all types of day studied. The average daily duration of weekly day was also calculated.

A description of the study sample was conducted: for quantitative variables, the mean and SD were calculated, and for qualitative variables, frequency and proportion were determined. To assess sex differences in all studied variables, the chi-square test was employed for qualitative variables, and the Student's t-test was used for quantitative variables if they followed a normal distribution, or the Mann-Whitney U test in case of non-normality. To assess the differences between central obesity status and the highest level of parental education or school status, the chi-square test was employed.

ROC curves (Receiver Operating Characteristic curves) were utilized to select MVPA and VPA thresholds associated with central obesity. The optimal threshold was determined based on the Youden index (J = sensitivity + specificity − 1) (*Perkins & Schisterman, 2006*).

Four different physical activity recommendations were considered: 1) WHO recommendation for MVPA: 60 minutes of daily average (RecMVPA-WHO). 2) Recommendation of 15 minutes of daily average of VPA from other studies (RecVPA-15) (*Martinez-Gomez et al., 2010*; *Füssenich et al., 2016*; *Schwarzfischer et al., 2017*). 3) MVPA recommendation obtained in this study through ROC curves (RecMVPA). 4) VPA recommendation obtained in this study through ROC curves (RecVPA). The percentage of children meeting all four physical activity recommendations was calculated.
The chi-square test was used to evaluate differences in the percentage of compliance with physical activity recommendations between children with and without central obesity. Likewise, it was assessed whether these four recommendations were equally associated with central obesity in boys and girls using the chi-square test.

The association between meeting recommendations and central obesity was assessed using the Phi correlation coefficient for binary variables, while the Cramer's contingency coefficient was used to comprehend the association of meeting recommendations with overweight and obesity.

The Mann-Whitney U test was used to assess differences in physical activity times between boys and girls, and between children with and without central obesity.
The Wilcoxon signed-rank test was used to assess differences in physical activity times between the three types of days studied.

The chi-square test was employed to evaluate the association between the percentage of compliance with physical activity recommendations and sex and central obesity. To assess differences in the percentage of compliance with physical activity recommendations between the three types of days studied, the McNemar test was used.

Conditional logistic regression models were used to calculate the odds ratios (OR) for compliance with recommendations on weekend days and school days with physical education class in students who met recommendations on school days without physical education class. Separate models were created for boys and girls and for those with or without central obesity. Additionally, adjustments were made for the highest level of education of the parents and school status. For all analyses, a significance level of $p < 0.05$ was established. Statistical analysis was performed using IBM® SPSS® Statistics version 25 for macOS (IBM Software Group, Chicago, IL, USA), except for conditional logistic regression, which was conducted using R (*R Core Team, 2021*).

## RESULTS

A total of 33 schools agreed to participate in the present study, out of the 40 schools invited. 22 groups of 3[rd] grade primary school students were evaluated between October 2019 and March 2020 when primary education schools were closed due to the onset of the COVID-19 pandemic. Therefore, 11 primary education schools were not assessed due to the pandemic.

A total of 510 informed consents were distributed in the 22 evaluated schools (an average of 23.2 children per classroom). 401 (78.6%) consents were correctly filled out and returned, of which, 385 children accepted to participate (75.5% of the total invited). Seven of the children authorized to participate were absent from class on the first day of assessment and could not be evaluated. In two assessments, data recorded on the accelerometer could not be extracted. No participants were excluded for being over 9 years old or having limitations in participating in physical activity during the assessment. Therefore, the sample with data derived from accelerometer assessments consisted of 376 children (73.7% of the total invited and 97.7% of the total accepted), of which, 360 children had at least four valid days, including at least two school days and one weekend day (Fig. 1).

In Table 1, the statistics of valid evaluated days are described.

In Table 2, descriptive statistics are presented for all participants together and for boys and girls separately. It includes the weekly average of physical activity at different intensities, as well as the weight status of children based on WHO and IOTF criteria. The proportion of children with central obesity is also shown. Daily average times for MPA, VPA, and MVPA were significantly higher in boys than girls, while LPA was higher in girls. Additionally, children with central obesity engaged in an average daily time of 77.2 minutes (SD, 27.4) of MVPA and 12.8 minutes (SD, 8.3) of VPA, significantly lower than those without central obesity, who averaged 88.6 minutes (SD, 28.9) of MVPA and 16.2 minutes (SD, 9.0) of VPA.

In addition, it was found that the highest level of parental education was related to central obesity status (with central obesity; without university studies: 35.0%; with university studies: 14.1%; $p < 0.001$). However, the school status did not show a significant relationship with central obesity (with central obesity; public school: 26.0%; private school: 25.3%; $p = 0.899$).

The threshold for the average daily time of MVPA below which it is associated with obesity was 75 minutes (76.4 in boys and 73.8 in girls). The threshold for the average daily time of VPA that is associated with obesity varied significantly by sex, thus both calculated values were used, 12.5 minutes in boys and 9.7 (rounded to 9.5) minutes in girls (Table 3). These values were employed as recommendations for average daily time of MVPA (RecMVPA) and VPA (RecVPA).

In Table 4, the percentages of compliance with different recommendations are observed, according to sex, and central obesity status. Compliance with RecMVPA-WHO (*i.e.*, 60 minutes on average daily MVPA), RecMVPA (*i.e.*, 75 minutes on average daily MVPA) and RecVPA (*i.e.*, 12.5 minutes in boys and 9.5 minutes in girls on average daily VPA) were statistically significantly associated with central obesity status. However, RecVPA-15 (*i.e.*, 15 minutes on average daily VPA) was not significantly associated with central obesity in girls, only in boys.

In the total number of participants with WHtR and BMI data ($n = 326$), a negative correlation was observed between compliance with the recommendations and central obesity and excess weight (overweight and obesity). Specifically, the following results were observed for the strength of the association with central obesity (RecMVPA-WHO (Phi

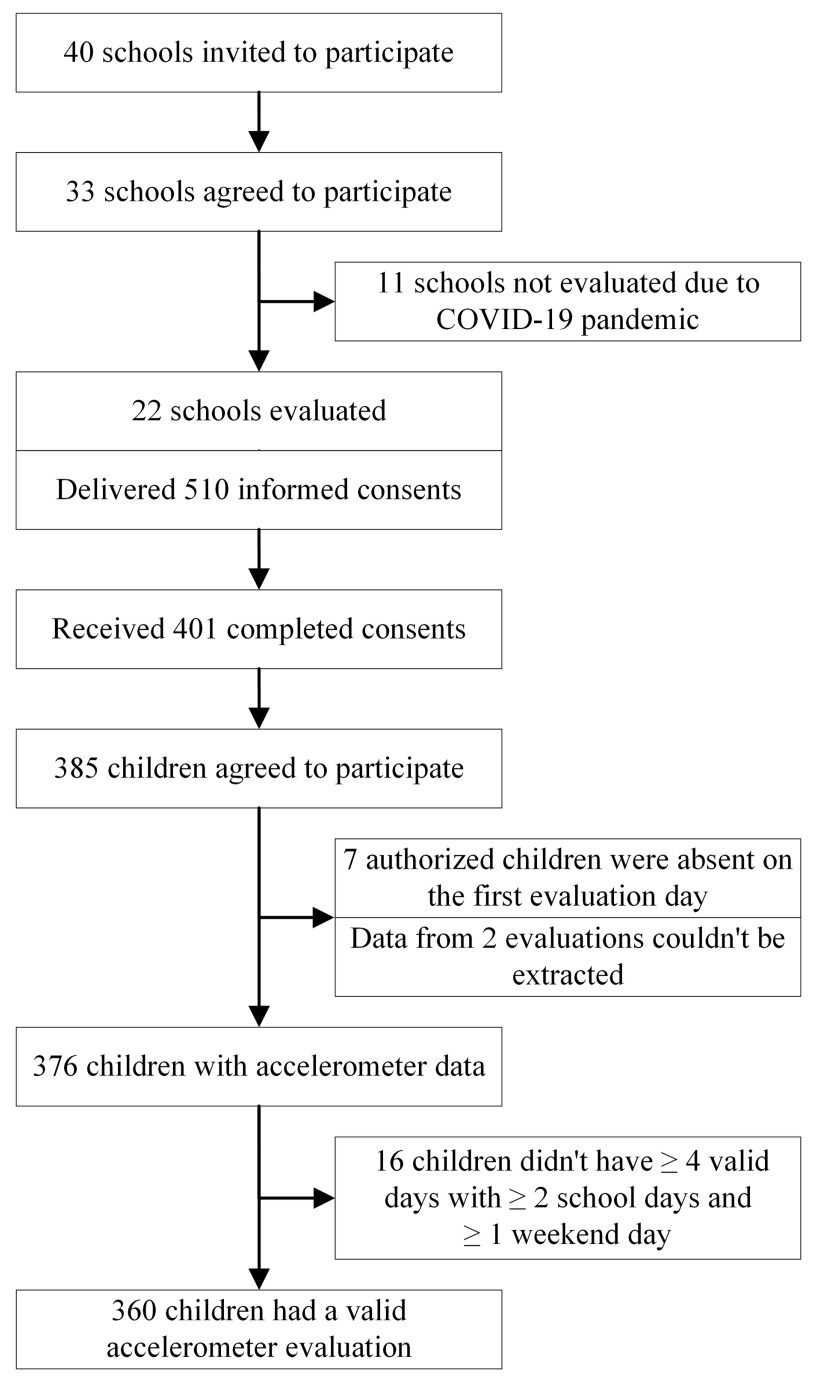

**Figure 1 Flowchart of participants' physical activity evaluation.**

coefficient, $p$ value): $-0.181$, $p = 0.001$; RecMVPA: $-0.213$, $p < 0.001$; RecVPA-15: $-0.111$, $p = 0.045$; RecVPA: $-0.247$, $p < 0.001$) and with excess weight (RecMVPA-WHO (Cramer's contingency coefficient, $p$ value): $0.142$, $p = 0.036$; RecMVPA: $0.141$, $p = 0.037$; RecVPA-15: $0.109$, $p = 0.142$; RecVPA: $0.174$, $p = 0.006$).

**Table 1 Descriptive statistics of types of day.**

| Type of day | Participants | Minimum | Maximum | Total days | Mean | SD |
|---|---|---|---|---|---|---|
| All days | 360 | 4 | 11 | 2,782 | 7.7 | 1.4 |
| Average weekly days | 360 | 4 | 9 | 2,502 | 7.0 | 1.1 |
| School days | 360 | 2 | 5 | 1,652 | 4.6 | 0.6 |
| School days with PEC | 357 | 1 | 3 | 689 | 1.9 | 0.7 |
| School days without PEC | 358 | 1 | 4 | 964 | 2.7 | 0.7 |
| Weekend days | 360 | 1 | 4 | 850 | 2.4 | 0.8 |
| Holiday days | 126 | 1 | 2 | 229 | 1.8 | 0.4 |
| School absence days | 42 | 1 | 3 | 50 | 1.2 | 0.6 |

Note:
Min, minimum; Max, maximum; SD, standard deviation; PEC, physical education classes.

**Table 2 Characteristics of participants according to study variables.**

| | | All participants $n = 360$ | | Boys $n = 176$ | | Girls $n = 184$ | | $p^{\dagger}$ |
|---|---|---|---|---|---|---|---|---|
| **Age (years)** | Mean, SD | 8.5 | 0.4 | 8.5 | 0.4 | 8.6 | 0.4 | 0.410 |
| **Waist (cm)** | | 61.4 | 8.9 | 62.3 | 9.4 | 60.7 | 8.3 | 0.093 |
| **Weight (kg)** | | 32.2 | 8.5 | 32.8 | 8.7 | 31.7 | 8.4 | 0.181 |
| **Height (cm)** | | 131.4 | 6.3 | 132 | 6.3 | 130.9 | 6.3 | 0.109 |
| **WHtR** | | 0.47 | 0.06 | 0.47 | 0.06 | 0.46 | 0.05 | 0.250 |
| **BMI (kg/m$^2$)** | | 18.4 | 3.7 | 18.6 | 3.7 | 18.3 | 3.6 | 0.337 |
| **Wear time (hours)** | | 24 | 0.3 | 23.9 | 0.3 | 24 | 0.4 | 0.102 |
| **Weekly PA (min/day)** | | | | | | | | |
| LPA | | 218.2 | 33.4 | 212.5 | 33.3 | 223.6 | 32.7 | **0.001** |
| MPA | | 70.6 | 21.5 | 76.8 | 23 | 64.7 | 18 | **<0.001** |
| VPA | | 15.6 | 9.2 | 19.9 | 10.5 | 11.4 | 5.1 | **<0.001** |
| LMVPA | | 304.7 | 52.4 | 309.7 | 55.4 | 299.9 | 49 | 0.075 |
| MVPA | | 86,5 | 29.1 | 97.2 | 31.7 | 76.3 | 21.9 | **<0.001** |
| **Central obesity (WHtR ≥ 0.50)** | $n$, % | 84 | 25.8 | 44 | 28.2 | 40 | 23.5 | 0.335 |
| **Children's weight status (WHO)** | | | | | | | | |
| Normal weight | | 160 | 49.1 | 71 | 45.5 | 89 | 52.4 | 0.349 |
| Overweight | | 79 | 24.2 | 38 | 24.4 | 41 | 24.1 | |
| Obesity | | 87 | 26.7 | 47 | 30.1 | 40 | 23.5 | |
| **Children's weight status (IOTF)** | | | | | | | | |
| Normal weight | | 192 | 58.9 | 90 | 57.7 | 102 | 60 | 0.868 |
| Overweight | | 77 | 23.6 | 37 | 23.7 | 40 | 23.5 | |
| Obesity | | 57 | 17.5 | 29 | 18.6 | 28 | 16.5 | |
| **Parent's educational level** | | | | | | | | |
| Non-university | | 182 | 55.5 | 87 | 55.4 | 95 | 55.6 | 0.979 |
| University | | 146 | 44.5 | 70 | 44.6 | 76 | 44.4 | |

(continued)

| | | All participants | | Boys | | Girls | | |
|---|---|---|---|---|---|---|---|---|
| | | $n$ = 360 | | $n$ = 176 | | $n$ = 184 | | $p^{\dagger}$ |
| **School status** | | | | | | | | |
| Public | | 265 | 73.6 | 128 | 72.7 | 137 | 74.5 | 0.710 |
| Private | | 95 | 26.4 | 48 | 27.3 | 47 | 25.5 | |

Notes:
$n$, number of participants; SD, standard deviation.
[†] $p$ value for the difference between boys and girls (Mann-Whitney U test, t-test or chi-square), $p < 0.05$ in bold; WHtR, waist-to-height ratio; BMI, body mass index; L, light; M, moderate; V, vigorous; PA, physical activity.

**Table 3 Results of ROC curve analyses for the associations among MVPA, VPA, and central obesity in 324 children aged 8- to 9-years.**

| | $n$ | AUC | 95% CI | | $p$ | Youden index | Threshold (min/day) | Sensitivity | Specificity |
|---|---|---|---|---|---|---|---|---|---|
| | | | Lower | Upper | | | | | |
| **MVPA** | | | | | | | | | |
| **Boys** | 156 | 0.64 | 0.54 | 0.74 | **0.008** | 0.231 | 76.4 | 77% | 46% |
| **Girls** | 168 | 0.64 | 0.55 | 0.74 | **0.006** | 0.307 | 73.8 | 59% | 71% |
| **VPA** | | | | | | | | | |
| **Boys** | 156 | 0.68 | 0.58 | 0.78 | **<0.001** | 0.298 | 12.5 | 83% | 47% |
| **Girls** | 168 | 0.64 | 0.54 | 0.74 | **0.007** | 0.283 | 9.7 | 65% | 63% |

Note:
MVPA, moderate-vigorous physical activity; VPA, vigorous physical activity; $n$, number of participants; AUC, area under the curve; CI, confidence interval; $p$ value for the AUC (ROC curve analyses), $p < 0.05$ in bold.

**Table 4 Percentage of compliance with different recommendations for MVPA and VPA, according to sex, and central obesity.**

| | All participants | | | Boys | | | Girls | | | |
|---|---|---|---|---|---|---|---|---|---|---|
| | $n$ = 360 | | | $n$ = 176 | | | $n$ = 184 | | | |
| | $n$ | % | $p^{*}$ | $n$ | % | $p^{*}$ | $n$ | % | $p^{*}$ | $p^{\dagger}$ |
| **RecMVPA-WHO** | 290 | 80.6 | | 153 | 86.9 | | 137 | 74.5 | | **0.003** |
| Central obesity | 56 | 66.7 | **0.001** | 32 | 72.7 | **0.006** | 24 | 60.0 | **0.027** | |
| Non-central obesity | 202 | 83.5 | | 101 | 90.2 | | 101 | 77.7 | | |
| **RecMVPA** | 229 | 63.6 | | 137 | 77.8 | | 92 | 50.0 | | **<0.001** |
| Central obesity | 37 | 44.0 | **<0.001** | 26 | 59.1 | **0.003** | 11 | 27.5 | **0.002** | |
| Non-central obesity | 164 | 67.8 | | 92 | 82.1 | | 72 | 55.4 | | |
| **RecVPA-15** | 151 | 41.9 | | 113 | 64.2 | | 38 | 20.7 | | **<0.001** |
| Central obesity | 27 | 32.1 | **0.045** | 21 | 47.7 | **0.004** | 6 | 15.0 | 0.420 | |
| Non-central obesity | 108 | 44.6 | | 81 | 72.3 | | 27 | 20.8 | | |
| **RecVPA** | 240 | 66.7 | | 131 | 74.4 | | 109 | 59.2 | | **0.002** |
| Central obesity | 39 | 46.4 | **<0.001** | 24 | 54.4 | **<0.001** | 15 | 37.5 | **0.002** | |
| Non-central obesity | 177 | 73.1 | | 92 | 82.1 | | 85 | 65.4 | | |

Notes:
MVPA, moderate-vigorous physical activity; VPA, vigorous physical activity; $n$, number of participants.
[†] $p$ value for the difference between boys and girls (chi-square), $p < 0.05$ in bold.
[*] $p$, value for the difference between children with and without central obesity (chi-square), $p < 0.05$ in bold; RecMVPA-WHO, MVPA (mean) $\geq$ 60 min/day; RecMVPA, MVPA (mean) $\geq$ 75 min/day; RecVPA-15, VPA (mean) $\geq$ 15 min/day; RecVPA, VPA (mean) $\geq$ 12.5 min/day in boys and $\geq$9.5 min/day in girls.

**Table 5** Minutes of average daily physical activity during physical education class (PEC) school days, non-PEC school days, and weekend days.

| | | | PEC days (1) | | | Non-PEC days (2) | | | Weekend days (3) | | | $p^{\dagger}$ | | |
|---|---|---|---|---|---|---|---|---|---|---|---|---|---|---|
| | | *n* | Mean | SD | $p^{*}$ | Mean | SD | $p^{*}$ | Mean | SD | $p^{*}$ | 1–2 | 1–3 | 2–3 |
| **Vigorous PA** | **All** | 355 | 22.0 | 12.8 | | 14.4 | 10.2 | | 11.0 | 9.6 | | <0.001 | <0.001 | <0.001 |
| | Boys | 171 | 28.0 | 14.3 | <0.001 | 18.8 | 12.0 | <0.001 | 14.3 | 11.7 | <0.001 | <0.001 | <0.001 | <0.001 |
| | Girls | 184 | 16.4 | 7.9 | | 10.4 | 6.0 | | 8.0 | 5.6 | | <0.001 | <0.001 | <0.001 |
| **All** | NCOb | 241 | 23.2 | 13.2 | <0.001 | 15.1 | 10.2 | 0.001 | 11.2 | 8.7 | 0.006 | <0.001 | <0.001 | <0.001 |
| | COb | 81 | 17.5 | 10.4 | | 11.5 | 8.6 | | 9.3 | 9.2 | | <0.001 | <0.001 | 0.001 |
| **Boys** | NCOb | 111 | 30.5 | 14.3 | <0.001 | 20.1 | 11.6 | <0.001 | 14.5 | 10.3 | 0.048 | <0.001 | <0.001 | <0.001 |
| | COb | 41 | 20.8 | 11.9 | | 14.3 | 10.4 | | 12.0 | 11.2 | | <0.001 | <0.001 | 0.017 |
| **Girls** | NCOb | 130 | 17.1 | 8.3 | 0.016 | 10.9 | 6.3 | 0.049 | 8.3 | 5.6 | 0.025 | <0.001 | <0.001 | <0.001 |
| | COb | 40 | 14.1 | 7.4 | | 8.7 | 4.8 | | 6.6 | 5.6 | | <0.001 | <0.001 | 0.024 |
| **Moderate-Vigorous PA** | **All** | 355 | 104.3 | 36.3 | | 81.7 | 31.3 | | 76.0 | 36.4 | | <0.001 | <0.001 | <0.001 |
| | Boys | 171 | 118.1 | 38.4 | <0.001 | 91.6 | 35.0 | <0.001 | 85.8 | 41.8 | <0.001 | <0.001 | <0.001 | 0.035 |
| | Girls | 184 | 91.5 | 28.8 | | 72.6 | 24.1 | | 66.9 | 27.7 | | <0.001 | <0.001 | 0.002 |
| **All** | NCOb | 241 | 107.8 | 37.9 | <0.001 | 83.7 | 31.1 | 0.002 | 77.7 | 34.4 | 0.003 | <0.001 | <0.001 | 0.005 |
| | COb | 81 | 91.6 | 30.9 | | 72.6 | 30.3 | | 66.7 | 34.8 | | <0.001 | <0.001 | 0.022 |
| **Boys** | NCOb | 111 | 125.2 | 39.2 | <0.001 | 94.9 | 33.9 | 0.005 | 87.1 | 37.8 | 0.108 | <0.001 | <0.001 | 0.063 |
| | COb | 41 | 98.7 | 33.0 | | 79.1 | 36.4 | | 76.9 | 42.2 | | <0.001 | 0.001 | 0.309 |
| **Girls** | NCOb | 130 | 93.0 | 29.7 | 0.034 | 74.2 | 25.1 | 0.052 | 69.7 | 29.0 | 0.005 | <0.001 | <0.001 | 0.029 |
| | COb | 40 | 84.3 | 27.1 | | 66.0 | 21.1 | | 56.2 | 21.1 | | <0.001 | <0.001 | 0.030 |

Notes:
PEC, physical education classes; n, number of participants; SD, standard deviation; PA, physical activity; NCOb, non-central obesity (waist-to-height ratio < 0.5); COb, central obesity (waist-to-height ratio ≥ 0.50)
[*] *p* value for the difference between categories (Mann-Whitney U test), *p* < 0.05 in bold.
[†] *p* value for the difference between (1) PEC school days, (2) non-PEC school days and (3) weekend days, in each category (Wilcoxon signed-rank test), *p* < 0.05 in bold.

No significant differences were found when comparing the average daily MVPA and VPA between children classified with central obesity and those classified with obesity. However, as can be observed in Table 5, significant differences were found in the performance of MVPA and VPA among all children classified with central obesity and non-central obesity on all types of days. However, no significant differences were found in the performance of MVPA on any type of day when comparing the three categories of weight status based on BMI. Differences in the performance of VPA were found based on weight status on school days with physical education class (*p* = 0.037) and without physical education class (*p* = 0.043), while none were found on weekends (*p* = 0.373).

In Table 6, the proportion of students who met different physical activity recommendations on different types of day is shown, by sex and by central obesity status. The compliance with the recommendation for VPA of 15 minutes daily on average is not shown, as it was found not to be associated with obesity in girls. The percentage of compliance with the recommendation for MVPA and VPA obtained in this study (75 minutes of MVPA, and 12.5 minutes in boys and 9.5 minutes in girls of VPA daily on average) was statistically significantly associated with central obesity in the entire sample on all three types of days evaluated.

**Table 6 Proportion of children who meet the physical activity recommendations on physical education classes (PEC) days, non-PEC days and weekend.**

| | | | PEC days (1) | | | Non-PEC days (2) | | | Weekend days (3) | | | $p^{\dagger}$ | | |
|---|---|---|---|---|---|---|---|---|---|---|---|---|---|---|
| | | | n | % | $p^*$ | n | % | $p^*$ | n | % | $p^*$ | 1–2 | 1–3 | 2–3 |
| **RecMVPA-WHO** | | **All** | 327/355 | 92.1 | | 266/355 | 74.9 | | 223/355 | 62.8 | | **<0.001** | **<0.001** | **<0.001** |
| | | Boys | 165/171 | 96.5 | **0.003** | 143/171 | 83.6 | **<0.001** | 118/171 | 69.0 | **0.020** | **<0.001** | **<0.001** | **<0.001** |
| | | Girls | 162/184 | 88.0 | | 123/184 | 66.9 | | 105/184 | 57.1 | | **<0.001** | **<0.001** | **0.013** |
| | All | NCOb | 223/241 | 92.5 | 0.306 | 188/241 | 78.0 | **0.002** | 158/241 | 65.6 | **0.017** | **<0.001** | **<0.001** | **<0.001** |
| | | COb | 72/81 | 88.9 | | 49/81 | 66.5 | | 41/81 | 50.6 | | **<0.001** | **<0.001** | 0.134 |
| | Boys | NCOb | 106/111 | 95.5 | 0.562 | 97/111 | 87.4 | **0.006** | 81/111 | 73.0 | 0.087 | **0.004** | **<0.001** | **0.005** |
| | | COb | 40/41 | 97.6 | | 28/41 | 68.3 | | 24/41 | 58.5 | | **<0.001** | **<0.001** | 0.289 |
| | Girls | NCOb | 117/130 | 90.0 | 0.093 | 91/130 | 70.0 | **0.041** | 77/130 | 59.2 | 0.063 | **<0.001** | **<0.001** | **0.020** |
| | | COb | 32/40 | 80.0 | | 21/40 | 52.5 | | 17/40 | 42.5 | | **0.019** | **0.001** | 0.424 |
| **RecMVPA** | | **All** | 282/355 | 79.4 | | 191/355 | 53.8 | | 164/355 | 46.2 | | **<0.001** | **<0.001** | **0.012** |
| | | Boys | 147/171 | 86.0 | **0.003** | 118/171 | 69.0 | **<0.001** | 95/171 | 55.6 | **0.001** | **<0.001** | **<0.001** | **0.002** |
| | | Girls | 135/184 | 73.4 | | 73/184 | 39.7 | | 69/184 | 37.5 | | **<0.001** | **<0.001** | 0.683 |
| | All | NCOb | 198/241 | 82.2 | **0.007** | 138/241 | 57.3 | **0.003** | 121/241 | 50.2 | **0.002** | **<0.001** | **<0.001** | 0.068 |
| | | COb | 55/81 | 67.9 | | 31/81 | 38.3 | | 25/81 | 30.9 | | **<0.001** | **<0.001** | 0.180 |
| | Boys | NCOb | 98/111 | 88.3 | 0.053 | 83/111 | 74.8 | **0.002** | 64/111 | 57.7 | 0.214 | **<0.001** | **<0.001** | **0.002** |
| | | COb | 31/41 | 75.6 | | 20/41 | 48.8 | | 19/41 | 46.3 | | **0.007** | **0.008** | >0.999 |
| | Girls | NCOb | 100/130 | 76.9 | **0.035** | 55/130 | 42.3 | 0.093 | 57/130 | 43.8 | **0.001** | **0.001** | **<0.001** | 0.878 |
| | | COb | 24/40 | 60.0 | | 11/40 | 27.5 | | 6/40 | 15.0 | | **0.001** | **<0.001** | 0.125 |
| **RecVPA** | | **All** | 308/355 | 86.8 | | 205/355 | 57.8 | | 138/355 | 38.9 | | **<0.001** | **<0.001** | **<0.001** |
| | | Boys | 153/171 | 89.5 | 0.146 | 117/171 | 68.4 | **<0.001** | 74/171 | 43.3 | 0.101 | **<0.001** | **<0.001** | **<0.001** |
| | | Girls | 155/184 | 84.2 | | 88/184 | 47.8 | | 64/184 | 34.8 | | **<0.001** | **<0.001** | **0.004** |
| | All | NCOb | 212/241 | 88.0 | **0.046** | 149/241 | 61.0 | **0.002** | 100/241 | 41.5 | **0.012** | **<0.001** | **<0.001** | **<0.001** |
| | | COb | 64/81 | 79.0 | | 34/81 | 42.0 | | 21/81 | 25.9 | | **<0.001** | **<0.001** | **0.011** |
| | Boys | NCOb | 101/111 | 91.0 | 0.075 | 83/111 | 74.8 | **0.002** | 51/111 | 45.9 | 0.192 | **<0.001** | **<0.001** | **<0.001** |
| | | COb | 33/41 | 80.5 | | 20/41 | 48.8 | | 14/41 | 34.1 | | **<0.001** | **<0.001** | 0.109 |
| | Girls | NCOb | 111/130 | 85.4 | 0.240 | 66/130 | 50.8 | 0.081 | 49/130 | 37.7 | **0.017** | **<0.001** | **<0.001** | **0.021** |
| | | COb | 31/40 | 77.5 | | 14/40 | 35.0 | | 7/40 | 17.5 | | **<0.001** | **<0.001** | 0.092 |

Notes:
MVPA, moderate-vigorous physical activity; VPA, vigorous physical; n, number of participants meeting the recommendations/all participants in that category.
$^*$ p value for the difference between categories (chi-square test), $p < 0.05$ in bold.
$^{\dagger}$ p value for the difference between (1) PEC school days, (2) non-PEC school days and (3) weekend days, in each category (McNemar test), $p < 0.05$ in bold;
RecMVPA-WHO, MVPA (mean) ≥ 60 min/day; RecMVPA, MVPA (mean) ≥ 75 min/day; RecVPA, VPA (mean) ≥ 12.5 min/day in boys and ≥ 9.5 min/day in girls;
NCOb, non-central obesity (waist-to-height ratio < 0.5); COb, central obesity (waist-to-height ratio ≥ 0.50).

In Fig. 2, the odds ratios for compliance with recommendations on school days with physical education class and on weekend days are shown for those children who complied with them on school days without physical education class, segmented by sex and central obesity. In the total participants, there was a higher likelihood of meeting the recommendations on school days with physical education class among those who already complied with them on school days without physical education class (RecMVPA-WHO (OR, 95% CI): 6.05, 4.26, 8.58; RecMVPA: 4.31, 3.23, 5.75; RecVPA: 5.47, 4.09, 7.32). However, it was less likely for children who met the recommendations on school days

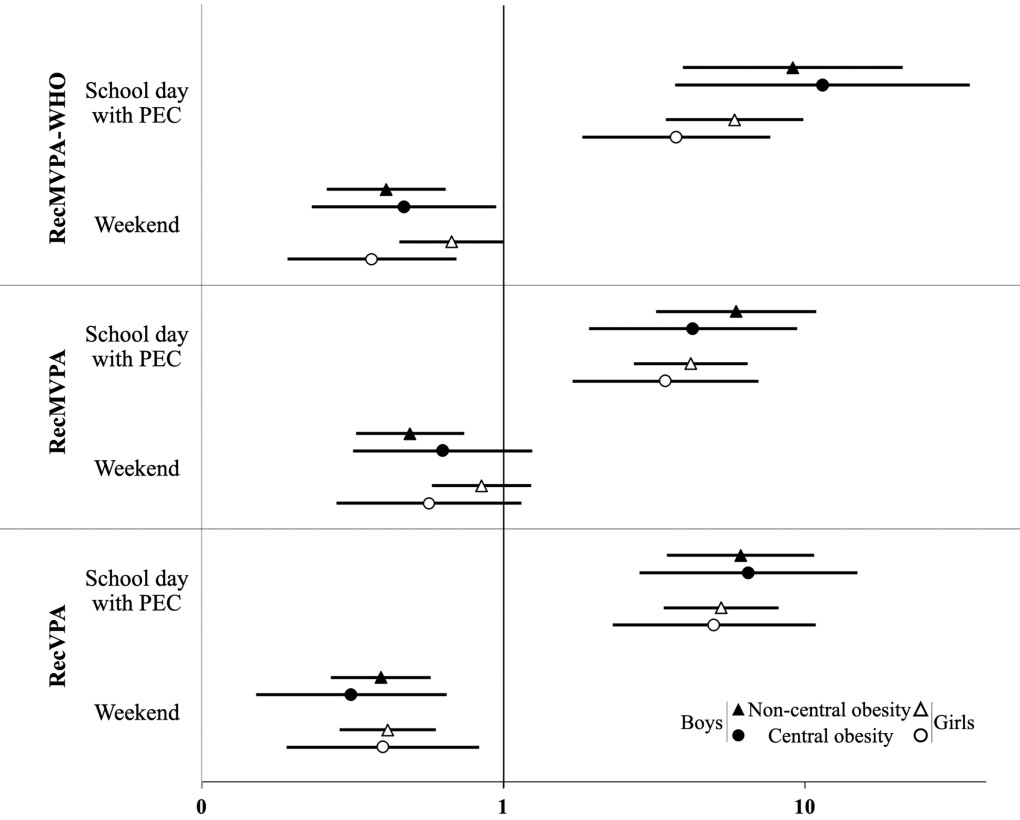

**Figure 2 OR and 95% CI for children who met the recommendations on school days without physical education clases (PEC, reference), to meet them on school days with PEC and on the weekend, by sex and central obesity.** MVPA, moderate-vigorous physical activity; VPA, vigorous physical activity; RecMVPA-WHO, MVPA (mean) > 60 min/day; RecMVPA, MVPA (mean) > 75 min/day; RecVPA, VPA (mean) > 2.5 min/day in boys and >9.5 min/day in girls; PEC, physical education class.

without physical education class to comply with them on weekend days (RecMVPA-WHO (OR, 95% CI): 0.50, 0.39, 0.64; RecMVPA: 0.63, 0.50, 0.80; RecVPA: 0.39, 0.31, 0.49).

# DISCUSSION

The results obtained in this study underscore differences in the physical activity and adherence to physical activity recommendations between school days with physical education class and the other types of day analyzed (school days without physical education class and weekend days). The use of specific physical activity thresholds to determine adherence to recommendations enhances the chances of promoting a healthy lifestyle and preventing health risks associated with physical inactivity. In accelerometer-based physical activity assessments, VPA emerges as a more sensitive tool for identifying associations between physical activity and the presence of obesity in children. As a result, replacing or complementing recommendations based on MVPA with those based on VPA can enhance physical activity and health measures. These findings emphasize the relevance of physical education classes and the importance of VPA in promoting healthy levels of physical activity in children.

Previous studies have reported low compliance with the WHO recommendation of achieving an average of at least 60 min of daily MVPA, both through questionnaires (*Aubert et al., 2018*; *Guthold et al., 2020*) and accelerometry (*Steene-Johannessen et al., 2020*). *Steene-Johannessen et al. (2020)* aggregated and harmonized accelerometry data from various studies conducted in European countries (including Spain), accumulating a sample of 47,497 participants. In this study, an average of 49.5 minutes of daily MVPA was reported in children, with a compliance rate with recommendations of 29% (95% CI [25–33]). The study integrated research conducted between 1997 and 2014. Data standardization required the use of a single axis and the reintegration of data into 60-s epochs. The impact of these methodological limitations on physical activity results was acknowledged by the study authors (*Steene-Johannessen et al., 2020*), and the low MVPA results and compliance with recommendations can be attributed, at least to some extent, to methodological decisions (*Kim et al., 2017*; *Llorente-Cantarero et al., 2021*; *Leppänen et al., 2022*). However, in our study, we have employed standardized protocols and data processing criteria recommended for the age and anatomical location of the accelerometer used in our sample (*Migueles et al., 2017*, *2019*), enabling for consistent comparisons with other studies that used similar approaches.

Other accelerometry-based studies have documented MVPA levels that are higher and consistent with our results (*Baquet et al., 2007*; *Roman-Viñas et al., 2016*; *Grao-Cruces et al., 2019*; *Camiletti-Moirón et al., 2020*; *Ávila-García et al., 2021*; *Schröder et al., 2021*; *Watson et al., 2023*). In an international sample of 6,128 children aged 9–11 years, daily accumulations of MVPA exceeding 60 minutes were found in six of the 12 countries included (*Roman-Viñas et al., 2016*). In a sample of 1,445 Spanish children and adolescents aged 6 to 17.9 years participants engaged in an average of 68.7 minutes (SD 25.6) of daily MVPA (*Camiletti-Moirón et al., 2020*).

Two previous studies were conducted in children residing in the same region (Andalusia, Spain) and with a similar age to the participants in this study. In one of the studies with a sample of 459 children, participants were found to engage in an average of 105.7 minutes (SD 27.9) of daily MVPA (*Ávila-García et al., 2021*). The other study, with a sample of 924 children, reported a daily average of 82.0 (SD 24.0) minutes in boys and 64.4 (SD ± 21.1) minutes in girls (*Grao-Cruces et al., 2019*). In the PASOS study (*Schröder et al., 2021*), an accelerometry assessment was conducted on 304 Spanish children and adolescents, resulting in 95.2 minutes (SD 33.2) of MVPA measured by accelerometry. In another study using the same accelerometer model as ours in a sample of 133 Australian children aged 9.4 years (SD 0.3), an average of 79.0 minutes (SD 28.0) of daily MVPA during the school year was observed (*Watson et al., 2023*). A study conducted on 26 French children aged 10.0 years (SD 1.0), revealed that participants accumulated 86.2 minutes of MVPA (*Baquet et al., 2007*). These results are positive, as maintaining high levels of physical activity carries health benefits (*Poitras et al., 2016*; *Chaput et al., 2020*).

It is important to highlight that, although no significant relationship was observed between the school status and our variables of interest, parental education level did show a significant relationship with central obesity status. These findings suggest that parental education level may play an important role in these aspects, especially the mother's

educational level (*van Ansem et al., 2014*), which deserves to be investigated in future studies.

The most notable discrepancies in the amount of physical activity performed at all intensity levels were observed between school days with physical education class and weekend days. Specifically, on average, children engaged in twice the amount of VPA and 37.2% more MVPA on school days with physical education class. Furthermore, differences were identified between school days with physical education class and those without physical education class (*i.e.*, 52.0% more VPA and 27.6% more MVPA on school days with physical education class). Additionally, the compliance with physical activity recommendations was higher during school days with physical education class compared to other types of day. These differences are primarily attributed to physical education classes, as our participants averaged 7.4 and 23.0 minutes per hour of VPA and MVPA, respectively, during their physical education classes (*Benavente-Marín et al., 2023*). Children who met the recommendations on school days without physical education class were more likely to meet them on school days with physical education class, with no significant differences based on sex or central obesity status. When comparing school days without physical education class to weekends, significant differences were also found in the accumulation of VPA (+30.8%) and MVPA (+7.5%), as well as a higher proportion of adherence to recommendations on school days without physical education class.

This finding is particularly relevant as higher-intensity physical activity has consistently shown a stronger association with health indicators than lower-intensity physical activity (*Füssenich et al., 2016*; *Poitras et al., 2016*; *Aadland et al., 2018*). Engaging in VPA and meeting physical activity recommendations, especially those based on VPA, can be a more effective strategy for preventing childhood obesity than strategies based on lower-intensity physical activity (*Gralla et al., 2016*; *Owens, Galloway & Gutin, 2016*; *García-Hermoso et al., 2021*).

Previous research supports these trends, reporting a significant increase in MVPA during school days with physical education class compared to school days without physical education class (*Meyer et al., 2013*; *Mooses et al., 2017*). Another study conducted in Spanish adolescents found higher levels of MVPA and greater adherence to WHO MVPA recommendation during school days with physical education class compared to school days without physical education class or weekend days (*Mayorga-Vega, Martínez-Baena & Viciana, 2018*). However, in contrast to our results, this latter study did not identify significant differences in adherence to the MVPA recommendation between days without physical education class and the weekend. These disparities could be due to the fact that they investigated the 2010 WHO MVPA recommendation (*i.e.*, at least 60 minutes of daily MVPA) (*World Health Organization (WHO), 2010*), or that their sample consisted of adolescents rather than children (*Van Hecke et al., 2016*). Our results also contrast with another study conducted in the Madrid region (Spain), which found higher levels of MVPA during the weekend than during the school days (*Laguna et al., 2013a*). Another possible explanation for the discrepancies found could be a different participation in organized extracurricular physical-sports activities. A total of 68.1% of the population of the present study declared to participate in organized extracurricular sports activities

(*Benavente-Marín et al., 2024*). Participation in this type of activities is not indicated in the studies by *Mayorga-Vega, Martínez-Baena & Viciana (2018)* or *Laguna et al. (2013a)*. But it is possible that, in more urban environments like Madrid, children may participate in a higher proportion in organized sports competitions during the weekends, which could result in higher levels of MVPA. Therefore, it is evident the need to know the participation in extracurricular activities to avoid possible biases when interpreting the results of average daily physical activity. Despite these divergences, the majority of accumulated evidence supports the findings of our study, indicating higher physical activity engagement and higher adherence to WHO MVPA recommendation during school days with physical education class compared to school days without physical education class and weekend days.

In our previous study we observed that there were no differences in declared participation in extracurricular sport activities between boys and girls (*Benavente-Marín et al., 2024*). Additionally, no differences in participation based on central obesity was observed. These findings suggest that participation in extracurricular sport activities is not influenced by either sex or central obesity. However, it is possible that children with central obesity and girls participate in less intense activities or might employ less intensity in their activities, which could explain the differences observed on days with physical education class. These aspects warrant further investigation in future studies to provide a more comprehensive understanding of the relationships between participation in extracurricular sport activities, physical activity, and obesity.

The relationship between physical activity on school days with and without physical education class and the cardiorespiratory fitness in Spanish children was investigated, concluding that an increase in the number of days with physical education class could raise adherence to the recommendations, regardless of cardiorespiratory fitness status (*Calahorro-Cañada et al., 2017*). Another study investigated the association between physical activity engagement and weekly physical education time, concluding that there was a positive association between children who had at least 60 minutes of weekly physical education with the amount of daily MVPA and VPA, compared to those with less than 60 minutes of weekly physical education classes (*Ikeda et al., 2022*). The importance of physical education classes in daily MVPA accumulation is highlighted in other studies (*Chen, Kim & Gao, 2014*; *Benavente-Marín et al., 2023*). In contrast, studies that examined the impact of interventions to increase physical activity in physical education classes concluded that differences in MVPA and VPA, while significant, were not clinically relevant (*Errisuriz et al., 2018*; *Huertas-Delgado et al., 2021*). Therefore, and in line with our results, an increase in the weekly frequency of days with physical education class appears to be a more cost-effective strategy than trying to increase the amount of physical activity performed during each physical education class to increase the daily amount of MVPA and VPA, as well as the likelihood of adherence to physical activity recommendations.

The correlation between specific volumes of daily VPA and health variables has been examined in some studies. *Schwarzfischer et al. (2017)* found that 15 to 20 minutes of VPA is comparable to 60 minutes of MVPA in reducing the risk of overweight, reinforcing the

recommendation of other authors to engage in at least 15 minutes of VPA per day (*Martinez-Gomez et al., 2010*; *Füssenich et al., 2016*; *Schwarzfischer et al., 2017*). Other studies have suggested a strong and favorable association between children who engage in an average of at least 10 minutes of daily VPA with adiposity, as well as cardiorespiratory fitness (*Gralla et al., 2016*; *García-Hermoso et al., 2021*). In our study, we found that at least 12.5 and 9.5 minutes of daily average VPA in boys and girls, respectively, were associated with the absence of central obesity. The positive relationship indicated by these studies between certain levels of VPA with health variables and the superior outcomes of VPA compared to lower-intensity physical activity are relevant factors to consider in optimizing future official recommendations. Therefore, a recommendation for daily average VPA could be clinically more relevant than the MVPA recommendation, as a public health program based on meeting a VPA recommendation requires less time investment, both for the intervened child or adolescent and for the entity or professional responsible for its implementation. Hence, complementing or replacing the MVPA recommendation with a VPA recommendation could enhance the effectiveness of this recommendation in optimizing the health of children and adolescents (*Martinez-Gomez et al., 2010*; *Gralla et al., 2016*; *Füssenich et al., 2016*; *Schwarzfischer et al., 2017*; *García-Hermoso et al., 2021*; *Gammon et al., 2022*).

Differences between boys and girls in physical activity and adherence to guidelines in children are consistently observed in most studies that present sex-segmented data (*Martinez-Gomez et al., 2010*; *Laguna et al., 2013b*; *Katzmarzyk et al., 2015*; *Füssenich et al., 2016*; *Corder et al., 2016*; *Schwarzfischer et al., 2017*; *Ferrer-Santos et al., 2021*). In line with previous evidence, significant differences between boys and girls were found in our study in all types of day studied, both in engaging in physical activity of all intensities and in meeting the MVPA recommendation. Only in the performance of LPA, girls outperformed boys. Moreover, it was observed that with higher physical activity intensity, there were statistically significant larger relative differences between sexes (boys *vs*. girls: LPA, −5.0%; MPA, +18.7%; VPA, +74.9%). For this reason, to study the level of compliance with physical activity recommendations, sex-segmented thresholds for MVPA and VPA associated with central obesity were calculated, in addition to using thresholds supported by previous evidence that do not distinguish between sexes (average MVPA ≥ 60 minutes/day; average VPA ≥ 15 minutes/day).

For the MVPA threshold, a minimal difference between boys and girls was found (*i.e.*, the MVPA threshold was 3.5% higher in boys), so a common threshold for both sexes of 75 minutes of daily average MVPA was chosen. However, for the VPA threshold, the difference between sexes was greater (*i.e.*, the VPA threshold was 28.9% higher in boys), so it was deemed appropriate to choose a different threshold for VPA for boys and girls (boys: 12.5; girls: 9.5 minutes/day of VPA). The RecMVPA-WHO, RecMVPA, and RecVPA showed a significant inverse association with central obesity, both in the total participants and in boys and girls separately. However, RecVPA-15 showed no significant association in girls (Table 3). For this reason, RecVPA-15 was excluded in analyses comparing recommendations between types of day. In fact, it seems that the higher the intensity of the

physical activity targeted by a recommendation, the greater the need to consider different thresholds for boys and girls for that recommendation.

The possibility of promoting different physical activity recommendations for boys and girls had already been raised by other authors (*Laguna et al., 2013b*; *Katzmarzyk et al., 2015*; *Gralla et al., 2016*; *Schwarzfischer et al., 2017*). One of this studies commented that the *World Health Organization (WHO)*'s *(2010)* recommendation on MVPA might be sufficient for girls to discriminate between those with normal weight and those with overweight or obesity. However, for boys, this recommendation was slightly lower than necessary to discriminate between those with normal weight and those with overweight or obesity (*Laguna et al., 2013b*). Similarly, another study concluded that more min of VPA and MVPA were needed in boys than in girls to find differences in the overweight status (*Schwarzfischer et al., 2017*). In two other studies, it was suggested that between 17–20 minutes of daily VPA in boys and between 9–11 minutes of daily VPA in girls mitigated the risk of overweight or obesity (*Katzmarzyk et al., 2015*; *Gralla et al., 2016*). In addition, an MVPA threshold for each sex was suggested in one of this studies (boys: 65 minutes/day 95% CI [55–75]; girls: 49 minutes/day 95% CI [43–62] (*Katzmarzyk et al., 2015*).

The thresholds proposed in our study differ from those commented previously (*Katzmarzyk et al., 2015*; *Gralla et al., 2016*), with higher and similar values for both sexes for MVPA and lower values with smaller differences between boys and girls for VPA. Studying the association between physical activity and health through accelerometry improves the accuracy of assessments, highlighting the need for specific thresholds for the methodology used and the target population. Furthermore, considering the overwhelming evidence of differences between boys and girls in physical activity, in line with our results, and the correlations with health variables found by authors who have proposed different recommendations for boys and girls, it seems appropriate to study physical activity recommendations with specific thresholds for boys and girls, with a particular emphasis on thresholds for VPA.

In the three types of day studied, significant differences were observed according to central obesity status in the performance of VPA in all participants, as well as in boys and girls separately. However, on the weekend, significant differences were found only in girls according to central obesity in the performance of MVPA. And, on school days without physical education class, significant differences were detected only in boys in the performance of MVPA. These results are consistent with previous evidence showing an inverse relationship between physical activity, especially VPA and MVPA, with excess weight or fat (*Hills, Andersen & Byrne, 2011*; *Gralla et al., 2016*; *Füssenich et al., 2016*; *Owens, Galloway & Gutin, 2016*; *World Health Organization (WHO), 2020*; *Chaput et al., 2020*; *García-Hermoso et al., 2021*).

On the other hand, on school days with physical education class, there were no significant differences in the compliance with RecMVPA-WHO based on central obesity or weight status. Differences were found in the compliance with RecMVPA and RecVPA in the total students based on central obesity status. Additionally, significant differences were found in the compliance with RecMVPA-WHO in the total participants based on central obesity status on school days without physical education class and on weekends,

while in the compliance with RecMVPA and RecVPA, significant differences based on central obesity status were found in boys on days without physical education class and in girls on weekends. Despite the differences found in the performance of MVPA and VPA on school days with physical education class, it seems that students manage to meet the recommendations studied on these days, reinforcing the recommendation to increase the weekly proportion of days with physical education class as a public health measure.

Other authors have shown a positive association between diet quality and an active lifestyle (*Fernández-Iglesias et al., 2021*). Thus, despite not having diet data in the present study, possibly diet quality and the results of physical activity assessed by accelerometry were positively related. This opens the door to studying physical activity assessed by accelerometry and diet quality in future studies.

Finally, the choice to focus on 8- and 9-year-old children is based on several factors. Firstly, this age group is at a developmental stage where significant changes have not yet occurred, allowing for a more accurate assessment of physical activity without the interference of developmental variability. Secondly, these children are mature enough to participate in more complex physical and sports activities. Lastly, this age group has been the subject of previous research in Spain, allowing for a direct comparison of our findings with previous studies.

The main strength of this study was the use of accelerometry as a tool for measuring children's physical activity. Unlike other subjective measurement methods, such as self-reported questionnaires, accelerometers have proven to be highly reliable and valid tools for assessing physical activity in children (*Sirard & Pate, 2001*; *Migueles et al., 2017*; *Gao et al., 2021*). Additionally, we obtained a very low non-compliance rate; 16 out of 376 participants (4%) with accelerometry data were not included in the analysis due to not having a sufficient number of valid days. Non-compliance rates have been recognized as one of the most important methodological limitations of accelerometry (*Howie & Straker, 2016*). Therefore, to prevent the non-compliance rate from being a limitation, we considered it vital to select an accelerometer model and an anatomical location that would optimize adherence to the assessment. The GENEActiv accelerometer placed on the wrist (*Fairclough et al., 2016*; *Leppänen et al., 2020*) allowed us to design a protocol in which it was not necessary to remove it at any time during the assessment, for activities such as water-related activities or sleeping. This optimized our compliance rate and transformed a potential limitation, as seen in most accelerometer studies, into a strength. Additionally, the GENEActiv accelerometer allowed us to obtain raw accelerometer data without any prior filtering through undisclosed licensed procedures. Hidden licensed filtering can bias the results, especially in the pediatric population, by underestimating intense activities (*Rowlands et al., 2016*; *Arvidsson, Fridolfsson & Börjesson, 2019*).

This study has some limitations that should also be acknowledged. The main limitation we encountered was the reduction in the evaluated sample compared to what we had planned to assess, due to the closure of primary education schools and home confinement resulting from the pandemic caused by the SARS-CoV-2 virus. Nevertheless, we managed to gather a sample that exceeds the predetermined minimum size needed to address our objectives. Another limitation caused by this exceptional situation was that we could not

assess children during the spring, which may bias our results, as there are more daylight hours and better weather conditions for engaging in physical activity during spring (*Remmers et al., 2017*; *Turrisi et al., 2021*). Additionally, there are inherent limitations in studying physical activity through accelerometry when comparing results from different studies, such as the accelerometer model, anatomical location, raw data processing methodology, with an emphasis on epoch duration, or selected cutoff points to determine physical activity intensity (*Baquet et al., 2007*; *Rowlands et al., 2016*; *Aadland et al., 2018*; *Arvidsson, Fridolfsson & Börjesson, 2019*; *Llorente-Cantarero et al., 2021*). Moreover, accelerometry does not adequately capture the intensity caused by physical activities such as cycling or strength training with resistance. Therefore, combining accelerometry with heart rate measurement may provide more accurate results (*Van Camp, Batchelder & Irwin Helvey, 2022*). Another limitation related to physical activity is that children participating in contact sports competitions were required by referees to remove accelerometers when competing. To try to reduce this limitation, we asked participants and their legal guardians to try to keep the accelerometer on, covering it with a wristband or bandage, provided the referee gave their approval. Nevertheless, some participants had to remove the accelerometer for their competitions. To address this bias, we studied the pattern of students who engaged in similar sports activities with the accelerometer on and imputed the periods of non-wear time when some participants reported having removed the accelerometer to engage in a known sport, whenever feasible. We did not evaluate the quality of the participants' diet or the use of the school canteen. Although we acknowledge that diet can significantly influence obesity and physical activity, we decided to focus on physical activity to keep the study design as unintrusive as possible for the participants and their families. Future research could benefit from the inclusion of these factors to provide a more comprehensive understanding of the relationships between diet, physical activity, and obesity. Another limitation of this study was that the anthropometric evaluation and the accelerometer evaluation were carried out separately, so in 34 participants (9.4% of the total) valid accelerometer data are available, but anthropometric results are not available. Finally, it should be noted that the observational nature of this cross-sectional study excludes any cause-and-effect association between physical activity or compliance with recommendations and sex, central obesity, excess weight, or their performance on different types of day.

## CONCLUSIONS

MVPA and VPA recommendations are presented for the prevention of obesity in children, specific to the study methodology. On days with physical education class, more physical activity was accumulated at all intensities than on days without this class, with the differences being greater at higher intensities of physical activity. In fact, VPA had a stronger correlation with the absence of obesity than lower-intensity activity. Adherence to the recommendations was also higher on days with physical education class. Therefore, increasing the weekly frequency of school days with physical education classes could be an effective strategy for preventing obesity in children. It was also observed that boys were physically more active and had higher adherence to the recommendations than girls.

## ACKNOWLEDGEMENTS

We thank the staff, pupils, parents, schools, and municipalities for their participation, enthusiasm, and support.

### Funding

This research was funded by Proyecto I+D+I Programa Operativo FEDER Andalucía 2014–2020 (UMA18-FEDERJA-114), and the University of Málaga. The funders had no role in study design, data collection and analysis, decision to publish, or preparation of the manuscript.

### Grant Disclosures

The following grant information was disclosed by the authors:
Proyecto I+D+I Programa Operativo FEDER Andalucía 2014–2020: UMA18-FEDERJA-114.
University of Málaga.

### Competing Interests

Julia Wärnberg and Napoleón Pérez-Farinós are Academic Editors for PeerJ.

### Author Contributions

- Juan Carlos Benavente-Marín conceived and designed the experiments, performed the experiments, analyzed the data, prepared figures and/or tables, authored or reviewed drafts of the article, and approved the final draft.
- Francisco Javier Barón-López analyzed the data, prepared figures and/or tables, authored or reviewed drafts of the article, and approved the final draft.
- Begoña Gil Barcenilla performed the experiments, authored or reviewed drafts of the article, and approved the final draft.
- Guadalupe Longo Abril performed the experiments, authored or reviewed drafts of the article, and approved the final draft.
- José M Rumbao Aguirre performed the experiments, authored or reviewed drafts of the article, and approved the final draft.
- Napoleón Pérez-Farinós analyzed the data, prepared figures and/or tables, authored or reviewed drafts of the article, and approved the final draft.
- Julia Wärnberg conceived and designed the experiments, analyzed the data, prepared figures and/or tables, authored or reviewed drafts of the article, and approved the final draft.

### Human Ethics

The following information was supplied relating to ethical approvals (i.e., approving body and any reference numbers):

The study was conducted in accordance with the Declaration of Helsinki, and approved by the research ethics committee CEI-Costa del Sol and the Portal de Ética de la

Investigación Biomédica de Andalucía-PEIBA, the 26th of September 2019, with the reference number 0114-2019.

## Data Availability

The data that support the findings of this study are restricted because the subjects are vulnerable children in schools. Sensitive data such as weight and height cannot be published even after deidentification given the small population that these subjects are drawn from. Permission to access the data can be requested from the ethics commission https://www.juntadeandalucia.es/salud/portaldeetica, with the following email address: portaldeetica.csalud@juntadeandalucia.es.

## Supplemental Information

Supplemental information for this article can be found online at http://dx.doi.org/10.7717/peerj.16990#supplemental-information.

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
