# Peer review of "Accelerometry-assessed daily physical activity and compliance with recommendations in Spanish children: importance of physical education classes and vigorous intensity"

_PeerJ, doi:10.7717/peerj.16990_

## Round 0.1 · original submission · Major Revisions

Both reviewers thought this was an interesting study, however, each suggested several points to improve the manuscript. Please pay particular attention to their suggestions for analysis. If you are not able to account for some of their concerns, consider noting it as a limitation.

·

Basic reporting

The article has a clear structure and is easy to read, although it would benefit from a re-reading by the authors to avoid repetitive content within the same sentence, as occurs in numerous sentences throughout the manuscript, as well as some grammatical and typographical errors. Tables and figures are clear and explained in detail. Specific questions about the data in one of the tables will be raised later. The authors have decided not to make their raw data available to the scientific community. In the interest of transparency and the essence of open science, it would be desirable that the raw data could be made available without the need for a request and eventual permission.
The background, methodological justification and discussion are solidly supported by a thorough review of recent literature. However, as will be indicated below, some bibliographical references are suggested that would help to improve some sections of the manuscript.
The authors do not explicitly state a hypothesis. It would be desirable to do so. In addition, they state too many objectives, which are not answered in the conclusions. It would be useful to set out a general objective and then break it down into specific objectives, for example, those set out in the current version. The conclusions should be rewritten according to this new structure, so that they respond to the objectives.

Experimental design

The research question is relevant, novel and clearly stated.
According to the information provided by the authors, the research has been conducted rigorously and according to the highest technical and ethical standards. This is particularly important given that the study has been carried out with children.
The methodology used is explained in detail and could be easily reproduced.

Validity of the findings

It is an interesting and relevant study, carried out in a rigorous manner. However, it suffers from some issues whose absence should be added or adequately justified, as they limit the validity of the study.

Firstly, the authors do not justify why they are studying 8-9 year olds and not other age groups. This justification needs to be included. The influence of age on exercise habits has been widely described, with a decrease in exercise practice between the ages of 11 and 15 years, much more marked among girls (see European Commission EU Action Plan on Childhood Obesity, available online). Moreover, in Spain, the highest prevalence of obesity among boys and girls is observed between 7 and 13 years of age, being higher in boys (Serra-Majem et al. 2003, Med Clin, doi: 10.1016/S0025-7753(03)74077-9). Given that the authors are aiming their study at prevention from a public health perspective, they have to justify the choice of this age group.

However, the main limitation is that eating habits are not considered at any point. Its evident relationship with obesity and their mutual influence on physical activity makes it essential to consider their influence. The authors should justify why they have not assessed the dietary habits of this population and analyse how the absence of control of these variables could be influencing the results they observe. In this sense, the authors should provide information on the use of the school canteen, which in many cases is related to children's participation in extracurricular activities and the design of whose menus is not under the control of the families.

Furthermore, the participation of children in out-of-school sports activities is not considered at any point. It would be necessary for the authors to indicate what proportion of the volunteers participate in after-school sports activities, as they could constitute a subgroup in themselves, both in terms of exercise habits, proportion of VPA and prevalence of overweight and obesity. It would therefore be necessary to know whether there are differences by gender or body composition in participation in extracurricular sports activities that could justify the differences observed by gender and body composition in the rest of the variables. In addition, participation in after-school sports activities usually involves participation in competitions at weekends, which could also condition the results observed at weekends.

In this respect, the statistically significant difference observed by the authors in VPA and MVPA between individuals with and without abdominal obesity, both on days with and without physical education classes and on weekends, is also striking. How can this difference be justified, particularly on days with physical education classes? It seems likely that participation in extracurricular sports activities is a determining factor in these differences. The authors should explore this question.

On the other hand, there are several issues that are raised but whose influence is not analysed. For example, the influence of parental education on the prevalence of overweight or obesity or on exercise habits is not studied. It has been extensively described that the educational background of the mother and not that of the father largely determines adherence to healthier eating habits (van Ansem et al. 2014, Int J Behav Nutr Phys Act, doi: 10.1186/s12966-014-0113-0). The different influence of the father's and mother's educational background on the variables of interest is not raised in the present study. The authors should consider this question. Furthermore, the influence of the type of school (public or private) is not explored either. In this sense, and linking with the questions raised previously, the availability of extracurricular sports activities in the school or competitive teams in different sports could condition the relationship between the type of school and physical activity, if it exists. The authors should explore this question further.

Finally, in general, throughout the manuscript, only aerobic exercise at different intensities is mentioned, but not the role of strength exercises. The authors should raise this limitation and explore it, given the increasing relevance of strength exercise in promoting population health at all ages.

Additional comments

INTRODUCTION
Throughout the introduction, cardiovascular risk in children is discussed. This does not seem very appropriate. It would be better to talk about the prevalence of cardiovascular risk factors or even the risk of metabolic pathologies (such as obesity or diabetes), the incidence of which is important in children. The prevalence of cardiovascular disease in childhood is extremely low and is usually independent of lifestyle. However, it is true that the risk of cardiovascular disease in adulthood is closely related to the presence of cardiovascular risk factors in childhood. Therefore, it seems more prudent to speak of prevalence of cardiometabolic risk factors rather than cardiovascular risk.

METHODS
The methods are clearly explained. However, there is repeated information on lines 192 and 220-221.

RESULTS
In Table 3, significant p-values are not highlighted in bold.

I do not understand in Table 6 the rows corresponding to All (n=355), Boys (n=171), and Girls (n=184). When subdividing by central obesity neither the total sum nor the sum by sex matches that number of subjects for both VPA and MVPA.

Furthermore, from the data in this table, the authors establish thresholds for VPA above which activity is not related to obesity prevalence in both boys and girls, but without considering intake. Are the authors suggesting that intake does not modulate obesity risk when aerobic physical activity is sufficiently long-lasting and intense? This is an interesting result, although perhaps daring given the absence of dietary records. In this regard, some authors have recently found that both healthy behaviours (i.e., an adequate diet, measured as adherence to the Mediterranean diet, and an active lifestyle), are associated with each other (Fernandez-Iglesias et al. 2021, Nutrients, doi: 10.3390/nu13051507). Perhaps this could justify what has been observed despite the lack of intake data.

DISCUSSION
Ln. 384. The term "performance" is confusing in the context in which it is used. It should be substituted.

Ln. 387-388 The authors state that using physical activity thresholds to determine adherence to exercise recommendations may improve diagnosis. Since the authors do not specify which pathology diagnosis they are referring to, perhaps the authors are guilty of over-interpretation of their results, which are very limited and come from a healthy population. Moreover, they do not consider any clinical variables beyond weight and body composition. It is therefore an empty sentence, not very specific and difficult to deduce from the available results, and should be deleted or reworded.

Ln. 389-390 Again, the authors overinterpret the results obtained, indicating that VPA is a sensitive tool for identifying the association between physical activity and health. The authors do not measure the health or disease status of the volunteer children, but only the presence of obesity and central obesity, as a single risk factor. Obesity is a very important risk factor, but its assessment alone does not determine the health status of an individual. Moreover, in the description of the sample selection, there is no indication of filtering by healthy population or by diseased population for certain pathologies. Again, there is an overestimation from a clinical perspective of the results obtained and the prudent thing to do is to rephrase or delete this sentence.

Ln. 457-469 The authors emphasise the observed difference in MVPA between weekdays without physical education classes and weekend days and compare this with other studies. A limitation of the present study is that participation in out-of-school physical activity is not recorded, as mentioned above. Typically, this activity involves competitions at weekends, which may be influencing the discrepant results across studies. As noted above, it would be valuable to incorporate this into the results and discussion. If this information is not available, it should be included in the discussion and highlighted as a limitation of the study.

Ln. 481-485 The interpretation given in these lines seems contradictory to that given in Table 5, where it is observed that, both in boys and girls, on weekdays without physical activity, on weekdays with physical activity and on weekends, those boys and girls without abdominal obesity participate in significantly more hours of VPA. How can the authors explain this discrepancy?

Ln. 494-495. The authors state that VPA can be a more effective intervention against childhood obesity than strategies based on lower-intensity physical activity. This statement should be rewritten to highlight the preventive value of VPA against obesity rather than its value as a treatment, as they do not take into consideration the difficulty of an obese person to perform efficiently, without risk of injury and with sufficient duration, VPA activities.

Ln. 496-499 These lines are lacking in cellular and molecular biology basis. They are extremely vague and dubious from a sound scientific perspective. They should be reworded or better deleted.

Ln. 513-517 The authors propose to incorporate VPA into exercise recommendations and as an intervention, even indicating that it should be carried out by certain professionals. However, the current recommendations in terms of MVPA are not part of mentored programmes, but are recommendations to be implemented autonomously by the population. Perhaps a mentored programme based on MVPA, so that the recommendations are covered, could be equally effective in the prevention of cardiometabolic risk factors in children, as the authors discuss in lines 395-434.
However, the difficulties of implementing a VPA intervention are not discussed, as it may not be available to all children given their physical condition or the presence of obesity, which exceeds 30% of the children in the sample. In this regard, it is interesting that the percentage of obese girls is 23% compared to 30% of boys, despite the fact that boys are more physically active in general and VPA in particular. This suggests that dietary habits are also playing an important role in the presence of obesity or that it may be necessary to correct the physical activity data measured by accelerometry for body weight, as suggested by some authors, such as Ekelund et al. 2003 (Am J Clin Nutr, doi: 10.1093/ajcn/79.5.851). Although no significant differences in body weight are observed between boys and girls, the mean weight of boys is about 3% higher than that of girls, which may be sufficient to affect physical activity measures.

CONCLUSIONS
The conclusion is extremely self-evident and seems to mask otherwise interesting results. Moreover, it does not meet the stated objectives and should be completely rewritten.

·

Basic reporting

The research named “The impact of physical education classes on adherence to daily physical activity recommendations among Spanish children aged 8-9, with emphasis on the role of vigorous intensity.“ revealed new insights children lifestyle habits and its importance in children health.
However, there are some questions to be solved. Data should be uploaded in an open science repository and some bibliography should also be considered such as Serra-Majem et al. 2003, Med Clin, doi: 10.1016/S0025-7753(03)74077-9 and Ansem et al. 2014, Int J Behav Nutr Phys Act, doi: 10.1186/s12966-014-0113-0

Experimental design

It was a research carried out rigorously and in accordance with the highest technical and ethical standards. Given that the study was conducted with children, this is especially significant. In this order, the methodology is clearly explained which opens the possibility of replication.

Validity of the findings

The main limitation is that diet patterns are not taken into account. It is crucial to take its influence into account on obesity because of their clear connection to it and their common impact on physical exercise. The authors should explain why they haven't evaluated this population's dietary preferences and examine how their inability to control these factors might affect the outcomes they see. Obesity groups could seem more active that non-obese ones, could this fact be related to diet pattern?
Another limitation of the study was that sports activities out of school classes were not considered. The authors should explain why and how this fact could modify the results observed.
Lastly, throughout the manuscript only aerobic exercise at different intensities is mentioned, but not the role of strength exercise. The authors should raise this limitation and explore it, given the increasing relevance of strength exercise in promoting population health at all ages.

Additional comments

Minor
Figure 2 title should be auto explained, some abbreviatures are not presented.
In table 2 all participants percentages of school status are wrong
In table 5 some of the numbers of participants don’t match between them when you have a part of a whole, 241+81≠355, 111+41≠171,130+40≠184, …

---

## Round 0.2 · accepted · Accept

The manuscript is ready for publication.

·

Basic reporting

I thank the authors for their kind and detailed replies. In my opinion, they have addressed and resolved all the issues raised in a proper and correct way. I am satisfied with the current version of the article, so I consider it, in its present form, ready for publication in Peer J.

Experimental design

No comment.

Validity of the findings

No comment.

·

Basic reporting

The report is well explained and the authors have modified the text according to the comments made in the first review.

Experimental design

No comment

Validity of the findings

No comment

Additional comments

No comment